# COVERAGE AND QUALITY DRIVEN TRAINING OF GENERATIVE IMAGE MODELS

ABSTRACT

Generative modeling of natural images has been extensively studied in recent years, yielding remarkable progress. Current state-of-the-art methods are either based on maximum likelihood estimation or adversarial training. Both methods have their own drawbacks, which are complementary in nature. The first leads to over-generalization as the maximum likelihood criterion encourages models to cover the support of the training data by heavily penalizing small masses assigned to training data. Simplifying assumptions in such models limits their capacity and makes them spill mass on unrealistic samples. The second leads to mode-dropping since adversarial training encourages high quality samples from the model, but only indirectly enforces diversity among the samples. To overcome these drawbacks we make two contributions. First, we propose a novel extension to the variational autoencoders model by using deterministic invertible transformation layers to map samples from the decoder to the image space. This induces correlations among the pixels given the latent variables, improving over commonly used factorial decoders. Second, we propose a training approach that leverages coverage and quality based criteria. Our models obtain likelihood scores competitive with state-of-the-art likelihood-based models, while achieving sample quality typical of adversarially trained networks.

## 1 INTRODUCTION

Natural images are notoriously difficult to model, yet it represents a fundamental problem of computer vision useful for many applications. Successful recent approaches can be divided into two broad families, which are trained in fundamentally different ways. The first is trained using likelihood-based criteria which ensure that all training data points are well covered by the model. This category includes variational autoencoders (Kingma & Welling, 2014; Kingma et al., 2016), autoregressive models such as PixelCNNs (van den Oord et al., 2016; Salimans et al., 2017), and flow-based models such as Real-NVP (Dinh et al., 2017). The second category is trained based on a signal that measures to what extent (statistics of) samples from the model can be distinguished from (statistics of) the training data, *i.e.* based on the quality of samples drawn from the model. This is the case for generative adversarial networks (GANs) (Goodfellow et al., 2014) and its numerous variants, as well as moment matching methods (Li et al., 2015).

Despite the phenomenal progress in this area in recent years, the existing method exhibit a number of drawbacks. Likelihood-based models are trained to put probability mass on all elements of the training set, however fitting all natural images perfectly would require infinite flexibility (even a finite, but reasonably big dataset is already too difficult to fit perfectly). Lack of flexibility forces models to over-generalize and assign probability mass on non-realistic images in order to cover all modes. Limiting factors in such models are the use of fully factorized decoders in variational autoencoders, restriction to the class of fully invertible functions in Real-NVP, and the lack of latent variables to induce global pixel dependencies in autoregressive models. Adversarial training on the other-hand explicitly ensures that the model does not generate samples that can be distinguished from training images. This goes, however, at the expense of covering all the training samples, a phenomenon known as "mode collapse" (Arjovsky et al., 2017). Moreover, adversarial trained models typically have a low-dimensional support, which prevents the use of likelihood to assess coverage of held-out data. Figure 1 gives and intuitive illustration of the opposite approaches of coverage and quality driven training.

To alleviate these drawbacks, we propose a model that uses adversarial mechanisms to explicitly discourage over-generalization together with maximum-likelihood training to avoid mode collapse. Our model is a novel extension of variational autoencoders (VAEs) that uses invertible transformations close to the output, thus avoiding naive independence assumptions on pixels given the latent

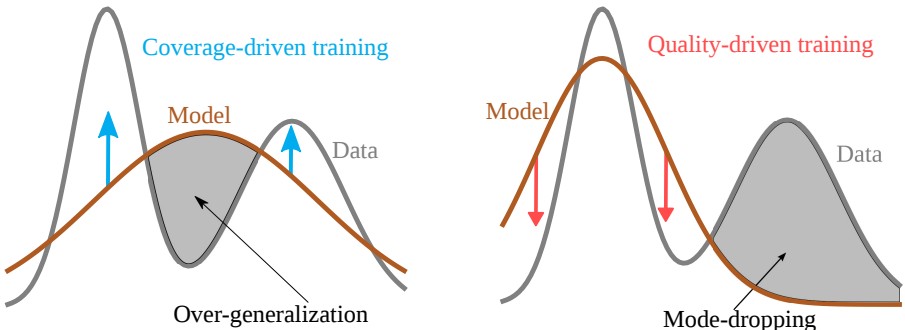

Figure 1: Intuitive explanation of the difference between coverage-driven and quality-driven training, in a one dimensional setting. CDT pulls probability mass towards points from regions of high density of the distribution underlying the data, while QDT pushes mass out of low-density regions. If the model is not flexible enough (in the example, it has too few modes), these training procedures lead to very different compromises in practice.

variables (which are typical in VAEs). As compared to Real-NVP, our model is computationally much more efficient, since most of the model consists of common non-invertible feed-forward layers.

To experimentally validate our coverage and quality driven training procedure and the different components of our model, we conduct a comprehensive and exhaustive set of experiments on the CIFAR-10 natural image dataset. We evaluate our models using likelihood scores on held-out training data, as well as inception score (IS) and Fréchet inception distance (FID). On a fixed architecture, our approach yields competitive likelihood performance while achieving at the same time a sample quality typical of GANs. We also provide additional qualitative and quantitative experimental results on the CelebA dataset, STL-10, ImageNet, and LSUN-Bedrooms, that further confirm the observations made on CIFAR-10.

## 2 PRELIMINARIES: COVERAGE VERSUS QUALITY DRIVEN TRAINING

The classic and most common approach to training generative models is to make sure that they cover the training data sampled from an unknown distribution $p^*$ well. Using a score $s_c(x, p_\theta)$ that evaluates how well a sample $x$ is covered by the model $p_\theta$, training proceeds by maximizing the objective $\mathcal{L}_C(p_\theta) = \int_{x \in X} p^*(x) s_c(x, p_\theta) \mathrm{d}x$. We refer to this training procedure as *coverage-driven training* (CDT). The other approach to generative modeling is to make sure that samples from the model fit the distribution $p^*$ underlying the training data. Given a score $s_q(x, p^*)$ that evaluates how plausible a sample $x$ is under $p^*$, a model $p_\theta$ can be trained by maximizing $\mathcal{L}_Q(p_\theta) = \int_{x \in X} p_\theta(x) s_q(x, p^*) \mathrm{d}x$. We refer to this procedure as *quality-driven training* (QDT). In practice $p^*$ is unavailable, and the integral over $p^*$ is therefore approximated using the empirical average over the training data. In what follows we argue that well-recognized failure modes of modern generative models, at least partially, stem from the choice of one of these procedures.

### 2.1 COVERAGE-DRIVEN TRAINING AND MAXIMUM LIKELIHOOD ESTIMATION

Among coverage-driven training methods, maximum likelihood estimation (MLE) is the most common. It maximizes the probability of data observed from $p^*$ under $p_\theta$ w.r.t. $\theta$, the parameter vector of the model, using the log-score $\mathcal{L}_C(p_\theta) = \mathbb{E}_{x \sim p^*}[\log p_\theta(x)]$. This is equivalent to minimizing $\mathrm{D}_{\mathrm{KL}}(p^* \parallel p_\theta)$, the Kullback-Liebler (KL) divergence between $p^*$ and $p_\theta$. This yields models that typically cover all the modes of the data, but tend to put mass in spurious regions of the target space; a phenomenon known as *over-generalization* (Bishop, 2006), and manifested by unrealistic samples in the context of generative image models. Intuitively, all the modes of the data are well covered because $p_\theta$ is explicitly optimized to cover all the samples from $p^*$. Conversely, what does not hap-

pen is sampling $x$ from $p_\theta$ and assessing its quality, ideally using the inaccessible $p^*(x)$ as a score. Therefore $p_\theta$ may put mass in spurious regions of the space without being heavily penalized.

Putting mass on samples from $p^*$ takes it away from spurious regions, so in principle an infinitely flexible model with infinite training data should fit $p^*$ without evaluating it on samples from $p_\theta$. Assuming $p^*$ is more complex than what $p_\theta$ can model, however, covering all the modes of $p^*$ will push $p_\theta$ to assign mass to regions of low probability under $p^*$. Even if $p_\theta$ were infinitely expressive, given a finite dataset it would over-fit by putting Diracs on all the training samples. The model is thus *forced to* put mass on points that are not in the training set, but we cannot control where this occurs, since $p^*$ cannot be evaluated. So regularization is necessary: if a simple model explains the data well it should generalize and spill some mass in the right places. Using a held-out validation or test set, we can assess if the model generalizes and assigns mass to the held-out samples from $p^*$, but not how much has been spilled elsewhere. Therefore, over-generalization is an inherent problem for MLE-based models, and having a mechanism that drives quality as well as coverage in the learned distribution is therefore desirable.

## 2.2 Adversarial models and quality-driven training

We have argued that models trained with maximum likelihood would benefit from having a mechanism that samples $x \sim p_\theta$ and evaluate it under $p^*$. Intuitively speaking, this is what happens when we ask human experts to assess a model by comparing its samples with their own implicit, expert approximation of $p^*$. One expects a good model $p_\theta$ to produce realistic samples, which is reasonably modeled by the idea that the cross-entropy between $p_\theta$ and $p^*$ should be small. One also expects the model to produce samples that are as diverse as possible, which is directly related to the entropy of $p_\theta$. Combining these ideas leads to the KL divergence $\mathrm{D}_{\mathrm{KL}}(p_\theta \,||\, p^*) = \mathbb{E}_{x \sim p_\theta}[\log p_\theta(x) - \log p^*(x)]$.

GANs use a similar mechanism: samples are drawn from the model and evaluated by a discriminator $D$. Originally, Goodfellow et al. (2014) trained the discriminator with the loss

$$\mathcal{L}_{\mathrm{GAN}} = \int_x p^*(x) \log D(x) + p_\theta(x) \log(1 - D(x))\mathrm{d}x, \tag{1}$$

and showed that, given $f(y) = a \log y + b \log(1 - y)$ is minimized for $y = \frac{a}{a+b}$, the optimal discriminator is given by

$$D^*(x) = \frac{p^*(x)}{p^*(x) + p_\theta(x)}. \tag{2}$$

Substituting the optimal discriminator, $\mathcal{L}_{\mathrm{GAN}}$ equals (up to additive and multiplicative constants) the Jensen-Shannon divergence $\mathrm{JSD}(p^*||p_\theta) = \frac{1}{2}\mathrm{D}_{\mathrm{KL}}(p^* \,||\, \frac{1}{2}(p_\theta + p^*)) + \frac{1}{2}\mathrm{D}_{\mathrm{KL}}(p_\theta \,||\, \frac{1}{2}(p_\theta + p^*))$. This loss, approximated by the discriminator, is symmetric and contains two KL divergence terms. While one of them is an integral on $p^*$ the term that approximates it, $\int_x p^*(x) \log D(x)$, is independent from the generative model. Therefore, it cannot be used to perform coverage-driven training, and instead, the generator is trained either to minimize $\log(1 - D(G(z)))$ or to maximize $\log D(G(z))$ (Goodfellow et al., 2014), where $G(z)$ is the deterministic generator that maps latent variables $z$ to the data space. Assuming $D = D^*$, the first of these generator objective functions becomes:

$$\log(1 - D^*(G(z))) = \int_x p_\theta(x) \log \frac{p_\theta(x)}{p_\theta(x) + p^*(x)} = \mathrm{D}_{\mathrm{KL}}(p_\theta \,||\, p_\theta + p^*). \tag{3}$$

As argued above, it seems reasonable that human assessment of generative models is also based on a quality-driven objective similar to that of the GAN generator, integrating a score function across samples from $p_\theta$. This could explain why images produced by GANs typically correlate well with human judgment. The main failure case of GANs, and more generally of quality-driven models, is that they typically do not cover the full support of the data. This well-recognized phenomenon is known as *mode-dropping* (Bishop, 2006; Arjovsky et al., 2017). GANs fail to provide an objective measure of this phenomenon, and of their performance in general. For instance, defining a valid likelihood requires adding volume to the low-dimensional manifold learned by GANs to define a density under which training and test data have non-zero density. Furthermore, computing the

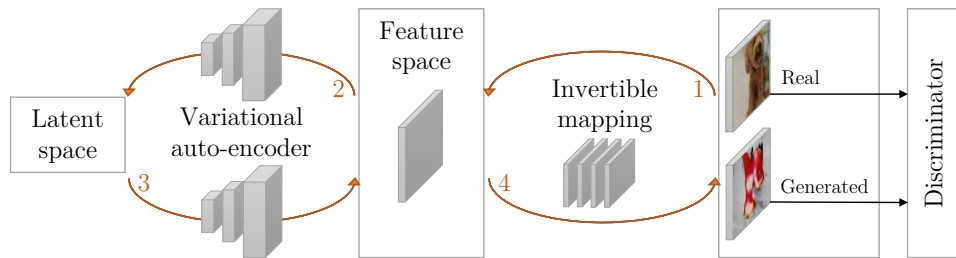

Figure 2: Schematic overview of our model. Our generative model is built on a VAE, with an additional invertible non-linear mapping $f(x)$ that maps $x$ to a feature space with the same dimension (shown with arrows 1, 4). The encoder takes $f(x)$ and maps it to a posterior distribution $q_\phi(z|x)$ over the latent variable (arrow 2), while the decoder maps $z$ to a distribution over the feature space (arrow 3). Defining the reconstruction loss in the feature space avoids the use of overly simplified per-pixel losses that are commonly used. The discriminator $D(x)$ is used to assess the quality of the samples from the generative model.

density of a datapoint under the defined density requires marginalizing out the latent variables. This is non trivial in the absence of a readily available inference model. Quality-driven training mirrors the shortcomings of coverage-driven training: the reverse KL divergence explicitly targets sample quality, and only implicitly how well the model covers the training data.

## 3 COVERAGE AND QUALITY DRIVEN TRAINING

Our analysis in the previous section motivates a generative learning framework that leverages both adversarial training and maximum likelihood estimation. In other words, we are interested in generative models with a non-degenerate support, where it is possible to assess generalization performance using likelihood measurements on held-out training data (CDT), and also provide high quality samples (QDT). Additionally, quality-driven training requires the generative model to allow for efficient sampling. This makes VAEs and flow-based models good candidates to build our approach on, as opposed to autoregressive models such as PixelCNNs which sample pixels sequentially. Between these two candidates, VAE have the added benefit that the decoder has a convolutional architecture similar to that of GAN generators. This allows the use of existing techniques to stabilize adversarial training.

VAEs rely on an inference network $q_\phi(z|x)$ to optimize a variational lower-bound on the data log-likelihood, often referred to as the evidence lower-bound (ELBO):

$$\text{ELBO}(x, \phi, \theta) = \mathop{\mathbb{E}}_{z \sim q_\phi(z|x)} \left[ \log(p_\theta(x|z)) - \text{D}_{\text{KL}}(q_\phi(z|x) \,||\, p_\theta(z)) \right] \le \log p_\theta(x). \qquad (4)$$

To ensure tractability of the computation of $\log p_\theta(x|z)$, typically strong independence assumptions are made on the components of $x$, given the latent variable $z$. For example, $p_\theta(x|z)$ is taken to be a fully factorized Gaussian, possibly with isotropic variance, *i.e.*, with covariance matrix $\Sigma = \sigma^2 I$. Clearly, such simplifying assumptions degrade modeling accuracy. In the case of natural image generation, this covariance structure corresponds to adding independent per-pixel noise to the image, which does not produce a realistic image, and leads the model to over-generalize to non-realistic examples. Yet, decreasing the "noise" level $\sigma \to 0$, pushes the model towards a degenerate low-dimensional support, which is heavily penalized by coverage-driven training. It is therefore important to relax this unrealistic independence assumption.

Carelessly lifting the independence assumption in Gaussian decoders $p_\theta(x|z)$ in VAEs requires the computation of intractable determinants. Using the decoder to predict a sparse Cholesky decomposition of the inverse covariance matrix allows for tractable non-factorizing Gaussian distributions (Dorta et al., 2018), and alleviates the situation to some extent. Flow-based models, however, offer a more flexible alternative that also allows departure from Gaussian, or other parametric, distributions. Models such as NVP (Dinh et al., 2017) map an image $x \in X$ from RGB space to a latent code $y \in Y$ using a bijection $f : X \to Y$, and rely on the change of variable formula to compute

the likelihood

$$p_X(x) = p_Y(f(x)) \left| \det \left( \frac{\partial f(x)}{\partial x^T} \right) \right|. \tag{5}$$

To sample $x$, we start with sampling $y$ from a parametric prior (typically a unit Gaussian), and use the reverse mapping $f^{-1}$ to find the corresponding $x$. Despite allowing for exact inference and efficient sampling, current flow-based approaches are worse than state-of-the-art likelihood-based approaches in terms of quantitative performance. They also lag behind adversarially trained models w.r.t. sample quality, while being more expensive to train.

We present a novel approach that builds on the benefits of these generative models as shown in Figure 2. In our model, we use an invertible function (implemented by NVP) to first map RGB images to an abstract feature space $f(x)$. A VAE is then trained to predict $f(x)$. This results in a non-factorial and non-parametric form of $p_\theta(x|z)$ in the space of RGB images. Although the likelihood of this model is intractable to compute, we can rely on a lower bound for training. The bound is obtained by combining the VAE variational lower bound (4), but defined in the feature space, with the change of variable formula (5) to account for the mapping between the feature and the RGB spaces, as follows:

$$\mathcal{L}_C(p_\theta) = - \mathop{\mathbb{E}}_{x \sim p^*} \left[ \text{ELBO}(f(x), \phi, \theta) + \log \left| \det \frac{\partial f(x)}{\partial x^T} \right| \right] \leq \mathop{\mathbb{E}}_{x \sim p^*} \left[ - \log p_\theta(x) \right]. \tag{6}$$

An alternative interpretation of our is model is to see it as a variant of NVP with a complex non-parametric prior distribution rather than a unit Gaussian. Our model combines benefits from VAE and NVP: it uses efficient non-invertible (convolutional) layers of VAE, while using a limited number of invertible layers as in NVP to avoid factorization of the conditional distribution $p_\theta(x|z)$. This is similar in spirit to the use of PixelCNNs as non-factorized decoders in VAEs (Chen et al., 2017; Gulrajani et al., 2017a; Lucas & Verbeek, 2018), but in addition allows for efficient sampling. The loss function $\mathcal{L}_C(p_\theta)$ enables coverage-driven training of this model.

To complement this with a quality-driven training objective, we rely on adversarial training. In particular, we use the modified objective proposed by Sønderby et al. (2017) that combines both generator losses considered by Goodfellow et al. (2014):

$$\mathcal{L}_Q(p_\theta) = - \mathop{\mathbb{E}}_{z \sim p_\theta(z)} \log \frac{D(G_\theta(z))}{1 - D(G_\theta(z))}. \tag{7}$$

Assuming the discriminator $D$ is trained to optimality at every step, it is easy to demonstrate that the generator is trained to optimize $\text{D}_{\text{KL}}(p_\theta \,||\, p^*)$, similar to our discussion in Section 2.2.

Adding the coverage (6) and quality (7) based loss functions, and assuming that the discriminator is trained to optimality at every step, the generator is explicitly trained to minimize a bound on the sum of two symmetric KL divergences,

$$\mathcal{L}_C(p_\theta) + \mathcal{L}_Q(p_\theta) \geq \text{D}_{\text{KL}}(p^* \,||\, p_\theta) + \text{D}_{\text{KL}}(p_\theta \,||\, p^*) + \text{H}(p^*), \tag{8}$$

where the entropy of the data generating distribution $\text{H}(p^*)$ is an additive constant that does not depend on the learned model. In our experiments we validate that this combined objective function enjoys the merits of both approaches, yielding models that produce convincing samples that are typical for GANs, and achieve likelihoods on held-out data that is comparable with existing models.

## 4 RELATED WORK

Some recent works are exploring models combining the benefits of auto-encoders and adversarial training. Larsen et al. (2016) use a VAE to model a feature space, taken to be one of the intermediate layers of a discriminator. This goes beyond the pixel-wise reconstruction loss in RGB space. However because the function going from pixel space to image space is not invertible this does not yield a density model in the RGB image space. In Rosca et al. (2017), a discriminator is used to estimate the likelihoods involved in the computation of the variational lower bound of VAEs. This combines GAN and VAE training mechansims, but does not yield a valid lower bound on the likelihood. Dumoulin et al. (2017) and Donahue et al. (2017) learn an encoder and decoder model using a discriminator that, given a pair $(x, z)$, predicts if $z$ was encoded from a real image, or if $x$ decoded

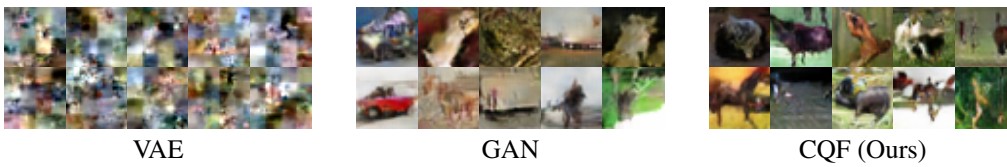

| VAE | GAN | CQF (Ours) |

Figure 3: Typical samples from the baseline GAN and VAE models, and our CQF model.

from a $z$ sampled from the prior. This procedure is fully adversarial and does not allow explicit optimization of a log-likelihood. This work is extended by Chen et al. (2018), which showed that it is possible to approximately optimize the symmetric KL in a fully adversarial setup and introduced reconstruction losses in RGB space to improve the correspondence between reconstructions and the ground truth. Ulyanov et al. (2018) collapse the encoder and a discriminator into one network that encodes real images and samples, and tries to separate their posteriors, yielding another fully adversarial approach. Makhzani et al. (2016) replace the regularization term on the latent variables of a VAE with a discriminator that compares latent codes from the prior and from the posterior. This regularization is more flexible, but does not lead to a valid density model on images.

As one of the main failure modes of GANs, mode-collapse has recently received considerable attention. One line of research is focused on allowing the discriminator to access batch statistics (Salimans et al., 2016; Lucas et al., 2018). Another line of research focuses on enforcing Lipschitz continuity of the discriminator (Arjovsky et al., 2017; Gulrajani et al., 2017b; Miyato et al., 2018). There are many recent advances in GANs that allow generation of high resolution samples when conditioning on image labels. Miyato & Koyama (2018) combine spectral normalization with a projection discriminator to yield convincing samples on ImageNet. Zhang et al. (2018) use a self-attention mechanism to allow the discriminator to better focus on salient long range dependencies. While conditional generation is outside of the scope of this paper, these contributions are orthogonal to our work and could be leveraged to further improve image quality. The same goes for the work of Karras et al. (2018), where the depth of the convolutional architectures used to build the two competing networks is progressively increased as training advances.

To go beyond the VAE independence assumption in pixel space, some recent works (Gulrajani et al., 2017a; Chen et al., 2017; Lucas & Verbeek, 2018) have proposed using autoregressive decoders. These models suffer from slow sampling because of their autoregressive components, and could not be directly improved with a quality-driven mechanism.

## 5 EXPERIMENTAL EVALUATION RESULTS

To evaluate our model and its variants (described below), we use the bits per dimension (BPD) measure, *i.e.*, negative log-likelihood normalized for the number of pixels and color channels, following the procedure from Dinh et al. (2017). We also report Fréchet inception distance (FID) (Heusel et al., 2017) and inception score (IS) (Salimans et al., 2016) measures, which are commonly used as a proxy to evaluate GAN models. We evaluate all measures using held-out data not used during training, which goes against common practice in many GAN papers that evaluate these measures using the training data. Implementation details can be found in Appendix A.

We begin with an ablation study on CIFAR-10, using a basic convolutional architecture with three convolutional layers in the generator, taken from Miyato et al. (2018), which shows that our approach yields the best of both worlds, and demonstrates the role and importance of each of its components. In Section 5.2 we study the addition of VAE and GAN refinements, and present results with models of increasing size to improve quantitative and qualitative performance. Then we explore different architectures of NVP in Section 5.3, and describe how the flexibility of the feature space built by the invertible mapping influences the overall performance of our model. We then compare to the state of the art on the CIFAR-10 and STL-10 datasets in Section 5.4, and present additional results at higher resolutions and on other datasets in Section 5.5. Note that all our models are trained in a fully unsupervised manner, *i.e.*, not conditioning on class labels or any other side information.

| | $\mathcal{L}_Q(p^*)$ | $\mathcal{L}_C(p^*)$ | Flow | BPD $\downarrow$ | IS $\uparrow$ | FID $\downarrow$ |
|---|---|---|---|---|---|---|
| GAN | ✓ | | | 7.0 | 6.8 | 31.4 |
| VAE | | ✓ | | 4.4 | 2.0 | 171.0 |
| VAEF | | ✓ | ✓ | 3.5 | 3.0 | 112.0 |
| CQ | ✓ | ✓ | | 4.4 | 5.1 | 58.6 |
| CQF | ✓ | ✓ | ✓ | 3.9 | 7.1 | 28.0 |

Table 1: Comparison of GAN and VAE baselines, our model using both losses (CQ), and our model using the additional flow-based layers (CQF).

| | IAF | Residual | BPD $\downarrow$ | IS $\uparrow$ | FID $\downarrow$ |
|---|---|---|---|---|---|
| GAN | | | — | 6.8 | 31.4 |
| GAN | | ✓ | — | 7.4 | 24.0 |
| CQF | | | 3.90 | 7.1 | 28.0 |
| CQF | | ✓ | 3.84 | 7.5 | 26.0 |
| CQF | ✓ | ✓ | 3.77 | 7.9 | 20.1 |
| CQF (large D) | ✓ | ✓ | 3.74 | 8.1 | 18.6 |

Table 2: Evaluation of more advanced architectures.

## 5.1 ABLATION STUDY ON CIFAR-10

We use 50k/10k train/test images of 32×32 pixels from the CIFAR-10 dataset (standard split). To evaluate our approach, we designed a baseline GAN architecture, with a few convolutional layers, which is stable and trains quickly. See Appendix A for details of the architecture. We train this adversarial baseline to optimize $\mathcal{L}_Q(p^*)$.

The generator is used as VAE decoder that produces the means of a factorizing Gaussian distribution over pixel RGB values, with isotropic covariance matrix $\sigma I$. We use a hierarchical set of latent variables as in (Sønderby et al., 2016; Bachman, 2016; Kingma et al., 2016), and the encoder (symmetric to the decoder) extracts deterministic features $h_i$ at different levels as the image is being encoded. While decoding the image, these features are used for top-down sampling and determine the posterior over latent variables at different depths in the decoder. This constitutes our VAE baseline, trained by optimizing $\mathcal{L}_C(p^*)$.

The combination of the encoder, decoder and discriminator is trained together to optimize $\mathcal{L}_C(p^*) + \mathcal{L}_Q(p^*)$. We refer to this model as CQ for "coverage and quality". We use a small invertible model to allow for a non-factorizing distribution across the pixels, which uses a single scale with three invertible layers, each composed of two residual blocks. We refer to our complete model as CQF, where the "F" denotes the addition of the flow-based component. The small invertible model in CQF increases the number of weights in the generator by roughly 1.4% so we also slightly increase the width of the generator in the CQ version for fair comparison. To endow the standard GAN generator with a full support and a valid likelihood on held-out test data, we add a trainable isotropic noise parameter $\sigma$ that does not depend on $z$, as in the VAE decoder. To approximate the log-likelihood we also add an inference network, which we train jointly with $\sigma$. Training is identical to that of the VAE model, except that the decoder parameters are frozen to the ones obtained using GAN training. Bear in mind that, as in a VAE setting, the inference network is imperfect and only allows us to compute a lower-bound on the likelihood.

The results in Table 1 show that a combination of the coverage and quality based loss functions of VAEs and GANs, results in a model that achieves significantly better samples, with IS increasing from 2.0 to 5.14 and FID dropping from 171.0 to 58.6, without sacrificing in terms of BPD. The gap in BPD performance between the CQ model and the GAN shows the superior coverage performance of the former. The addition of a small invertible model improves the model across all metrics, underlining the importance of a non-factorizing decoder.

| Scales | Blocks | BPD ↓ | IS ↑ | FID ↓ |
|--------|--------|-------|------|-------|
| 1 | 2 | 3.77 | **7.9** | **20.1** |
| 2 | 2 | 3.48 | 6.9 | 27.7 |
| 2 | 4 | **3.46** | 6.9 | 28.9 |
| 3 | 3 | 3.49 | 6.5 | 31.7 |

(a) CQF models

| Scales | Blocks | BPD ↓ | IS ↑ | FID ↓ |
|--------|--------|-------|------|-------|
| 1 | 2 | 3.52 | 3.0 | 112.0 |
| 2 | 2 | **3.41** | **4.5** | 85.5 |
| 3 | 2 | 3.45 | 4.4 | **78.7** |
| 4 | 1 | 3.49 | 4.1 | 82.4 |

(b) MLE models

Table 3: Evaluation on CIFAR-10 of different architectures of the invertible layers of the model.

In Figure 3 we show samples of the VAE and GAN baselines, as well as from our CQF model. VAE samples exhibit typical over-generalization behavior, which improves by using a deeper model but this issue remains noticeable. Our CQF model, on the other hand, produces samples with quality that is on par with that of the GAN samples, while also having reasonable likelihood, see Table 1, and therefore covering the support of the dataset well. In Appendix C we provide reconstructions qualitatively demonstrating the inference abilities of our CQF model. As is typical with expressive VAE model, training images and reconstructions are indistinguishable to the naked eye.

## 5.2 ARCHITECTURAL REFINEMENTS

To further improve quantitative and qualitative performance, we proceed to include two recent advances in the VAE and GAN literature. Gulrajani et al. (2017b) have shown a deeper discriminator with residual connections to be beneficial to training. We also increase the size of the generator to mirror these modifications. Kingma et al. (2016) improve VAE encoders by introducing inverse auto-regressive flow to allow for more accurate posterior approximations that go beyond factorized Gaussian approximations that are commonly used. These two upgrades are then used together to yield our final model.

The results presented in Table 2 improve monotonically in all metrics when adding residual connections and inverse auto-regressive flow (IAF in the table). Increasing the size of the discriminator (denoted "large D") yields further improvements with IS 8.1 and FID 18.6. This shows that our model can benefit from the architectural advances, which have been recently proposed for the component methods that we build on.

## 5.3 INFLUENCE OF THE FEATURE SPACE FLEXIBILITY

We now assess the impact of the expressiveness of the invertible model on the behavior of our framework. Popular invertible models such as NVP (Dinh et al., 2017) readily offer the possibility of extracting latent representation at several scales, separating global factors of variations from low level detail. We experiment with varying number of scales and residual blocks used in each invertible layer. Note that all the models evaluated so far in the previous sections are based on a single scale and two residual blocks. In addition to our CQF models, we also compare with similar models trained with maximum likelihood estimation (MLE).

The results in Table 3 show that factoring out features at two scales rather than one is helpful in terms of BPD. For the CQF models, however, the IS and FID scores deteriorate with more scales, and so a tradeoff between must be struck. For the MLE models, the visual quality of samples alsoimproves when using multiple scales, as reflected in better IS and FID scores. Their quality, however, remains far worse than those produced with the coverage and quality training used for the CQF models. Samples in the maximum-likelihood setting are provided in Appendix B. With three or more scales, models exhibit symptoms of overfitting: train BPD keeps decreasing while test BPD starts increasing, and IS and FID also degrade.

## 5.4 COMPARISON TO THE STATE OF THE ART

In Table 4 we compare the performance of our models with state-of-the-art generative image models. Many entries in the table are missing, since most adversarial methods are not able to assess likelihood of held-out data, and most likelihood-based models do not report IS or FID scores.

| | CIFAR-10 | | | STL-10 | | |
|---|---|---|---|---|---|---|
| | BPD ↓ | IS ↑ | FID ↓ | BPD ↓ | IS ↑ | FID ↓ |
| DCGAN (Radford et al., 2016) | | 6.6 | | | | |
| SNGAN (Miyato et al., 2018) | | 7.4 | 29.3 | | 8.3 | 53.1 |
| SNGAN-Hinge (Miyato et al., 2018) | | | | | 8.7 | 47.5 |
| BatchGAN (Lucas et al., 2018) | | 7.5 | 23.7 | | 8.7 | 51 |
| WGAN-GP (Gulrajani et al., 2017b) | | 7.9 | | | | |
| SNGAN-ResNet-Hinge (Miyato et al., 2018) | | **8.2** | 21.7 | | **9.1** | **40.1** |
| NVP (Dinh et al., 2017) | 3.49 | | | | | |
| VAE-IAF (Kingma et al., 2016) | 3.11 | | | | | |
| PixelRNN (van den Oord et al., 2016) | 3.00 | | | | | |
| PixelCNN++ (Salimans et al., 2017) | **2.92** | 5.5 | | | | |
| SVAE-r (Chen et al., 2018) | | 7.0 | | | | |
| CQF [+Residual, +flow, +large D] (Ours) | 3.74 | 8.1 | **18.6** | 4.0 | 8.6 | 52.7 |
| CQF [+Residual, +flow, +2 scales] (Ours) | 3.48 | 6.9 | 28.9 | **3.82** | 8.6 | 52.1 |

Table 4: Comparison of our models on CIFAR-10 and STL-10 (48×48) with state-of-the-art generative likelihood-based and adversarial models, as well as the hybrid SVAE.

On CIFAR-10 our best model achieves very competitive IS (on par with the best SNGAN version even though we do not use spectral normalization nor hinge loss) and the best FID. On STL-10 our model, trained using 100k/8k train/test images, also achieves very good IS and FID scores, losing only to SNGAN. To the best of our knowledge, we are the first to report BPD measurements on STL-10, and can therefore not compare to previous work in this metric. Figure 4 displays samples from our best models on these datasets.

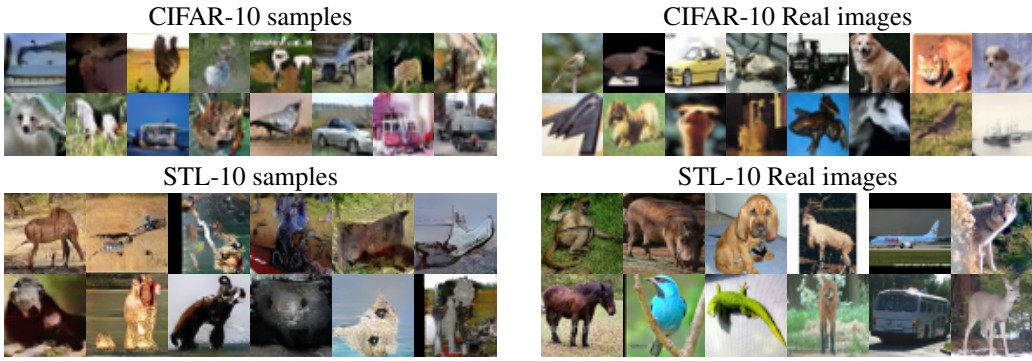

Figure 4: Random samples obtained from our best models, as reported in Section 5.4, on the CIFAR-10 and STL-10 (48 × 48) datasets.

## 5.5 RESULTS ON ADDITIONAL DATASETS

To further validate our approach we train our best models on images with higher resolutions, using our best models in terms of IS as reported in 2. For the CelebA dataset we used 196k/6.4k train/test images, resized to 96 × 96, and used central image crops of both 178 × 178 and 96 × 96 pixels. We also train on STL-10 (100k/8k train/test images) resized to 96 × 96, on the LSUN-bedrooms dataset (3M/300 train/test images) at 64 × 64 resolution, and on ImageNet (1.2M/50k train/test images) resized to 64 × 64 pixels.

We provide random samples and real images for these datasets in Figure 5, and quantitative evaluation results in Table 5. On the CelebA and LSUN datasets, our CQF generator is produces compelling samples despite the high resolution of the images. The samples for STL-10 and ImageNet are less realistic, due to the larger variability in these datasets (recall that we do not condition on

| | Resolution | BPD↓ | IS↑ | FID↓ |
|---|---|---|---|---|
| CelebA, crop 178 | $96 \times 96$ | 2.85 | — | 24.3 |
| CelebA, crop 96 | $96 \times 96$ | 2.45 | — | 13.8 |
| STL-10 | $96 \times 96$ | 3.85 | 8.8 | 100.8 |
| ImageNet | $64 \times 64$ | 4.90 | 7.6 | 69.9 |
| LSUN-Bedrooms | $64 \times 64$ | 4.01 | — | 61.9 |

Table 5: Evaluation of our CQF model on additional datasets. IS is not reported on CelebA and LSUN because it is not informative on these datasets.

class labels for generation). On CelebA, all scores improve significantly when using central crops of $96 \times 96$, due to the reduced variability in the smaller crop which removes the background from the images.

| Samples | Real images |
|---|---|

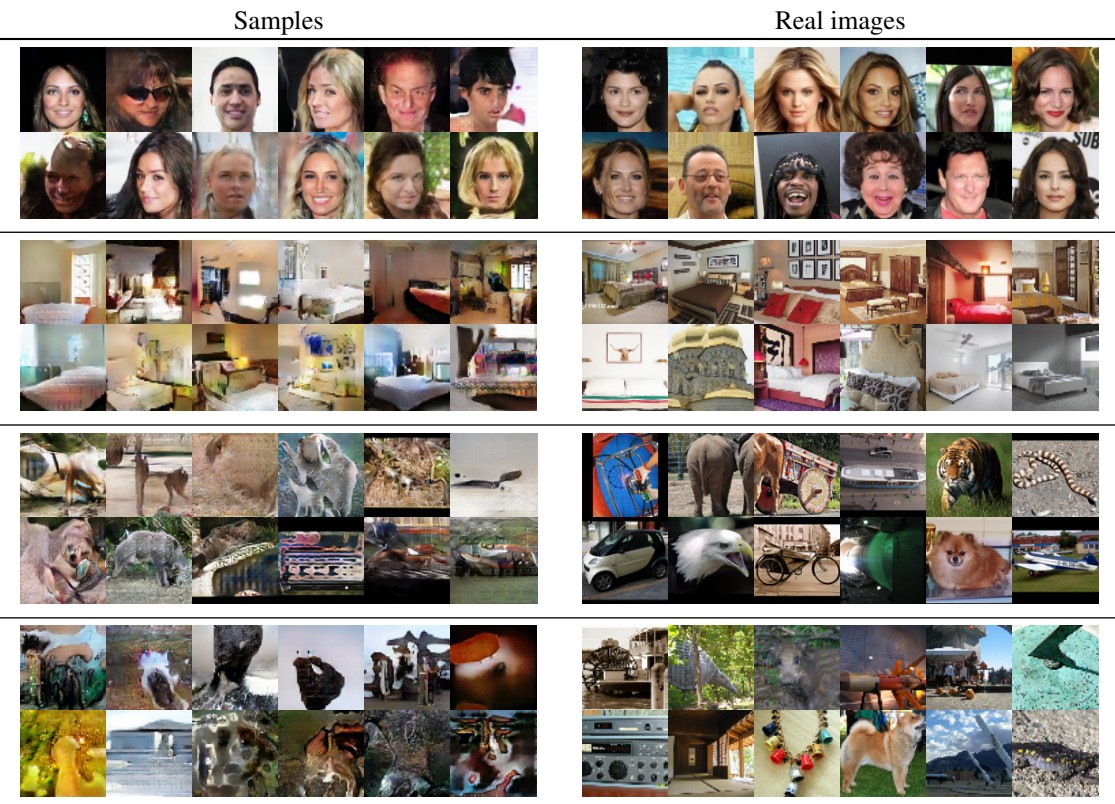

Figure 5: From top to bottom, random samples and real images from the CelebA (crop 178×178), LSUN-Bedrooms, STL-10 96×96, and ImageNet.

## 6 CONCLUSION

This paper presents a new generative model combining the two principal approaches to natural image generation: coverage and quality driven training. We also leverage invertible transformation layers to relax the conditional independence assumption in RGB space commonly made in VAE models. This allows us to achieve competitive results on many datasets, and also provide high quality samples typical of adversarial models, while keeping good likelihood scores on held-out data. Thus, providing a strong indication that our models do not suffer from mode collapse which is common in GAN models. We will release our code base upon publication.

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

## A    IMPLEMENTATION DETAILS

We used Adamax (Kingma & Ba, 2015) with learning rate 0.002, $\beta_1 = 0.9$, $\beta_2 = 0.999$ for all experiments. All CIFAR-10 experiments use batch size 64, other experiments in high resolution use batch size 32. To stabilize the adversarial training we use the gradient penalty (Gulrajani et al., 2017b) with coefficient 100, and 1 discriminator updates per generator update. We experimented with different weighting coefficient between the two loss components, and found that values in the range 10 to 100 on the adversarial component work best in practice. In this range, no significant inflence on the final performance of the model is observed, though the training dynamics in early training are improved with higher values. With values significantly smaller than 10, discriminator collapse was observed in a few isolate cases. All experiments reported here use coefficient 100.

For experiments with hierarchical latent variables we use 32 of them per layer. In the generator we use ELU nonlinearity, in discriminator with residual blocks we use ReLU while in simple convolutional discriminator we use leaky ReLU with slope 0.2.

Unless stated otherwise we use three NVP layers with a single scale and two residual blocks that we train only with the likelihood loss. Regardless of the number of scales, the VAE decoder always outputs a tensor of the same dimension as the target image, which is then fed to the NVP layers. Just like in reference implementations we use both batch normalization and weight normalization in NVP and only weight normalization in IAF.

We use reference implementations of IAF and NVP released by authors.

| Discriminator | Generator |
|---|---|
| | conv $3 \times 3$, 16 |
| conv $3 \times 3$, 16 | IAF block 32 |
| ResBlock 32 | IAF block down 64 |
| ResBlock down 64 | IAF block down 128 |
| ResBlock down 128 | IAF block down 256 |
| ResBlock down 256 | $h \sim \mathcal{N}(0; 1)$ |
| Average pooling | IAF block up 256 |
| dense 1 | IAF block up 128 |
| | IAF block up 64 |
| | IAF block 32 |
| | conv $3 \times 3$, 3 |

Table 6: Residual architectures for experiments from Table 2 and Table 3

## B QUALITATIVE INFLUENCE OF THE FEATURE SPACE FLEXIBILITY IN A MAXIMUM-LIKELIHOOD SETTING

In Figure 6 we show samples obtained using VAE models trained with MLE. The models include one without invertible decoder layers, and with NVP layers using one, two and three scales. The samples illustrate the dramatic impact of using invertible NVP layers in these autoencoders.

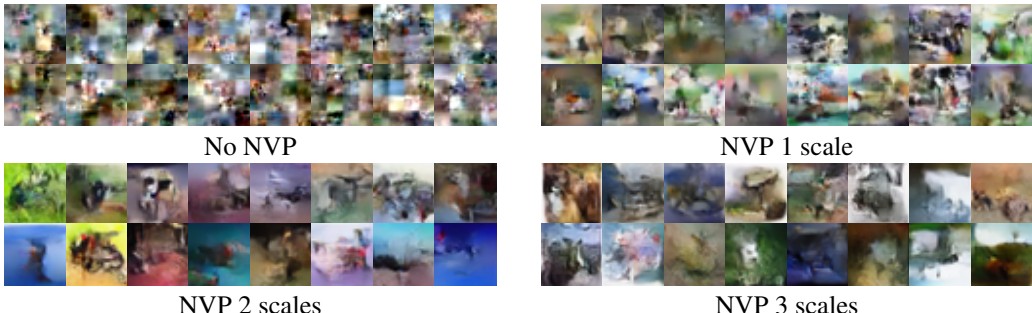

No NVP        NVP 1 scale

NVP 2 scales        NVP 3 scales

Figure 6: Samples from MLE models (Table 3b) showing qualitative influence of multi-scale feature space.

## C VISUALISATIONS OF RECONSTRUCTIONS

We display reconstructions obtained by encoding and then decoding ground truth images with our models (CQ and CQF from table 1) in Figure 7. As is typical for expressive Variational auto-encoders, real images and their reconstructions cannot be distinguished visually.

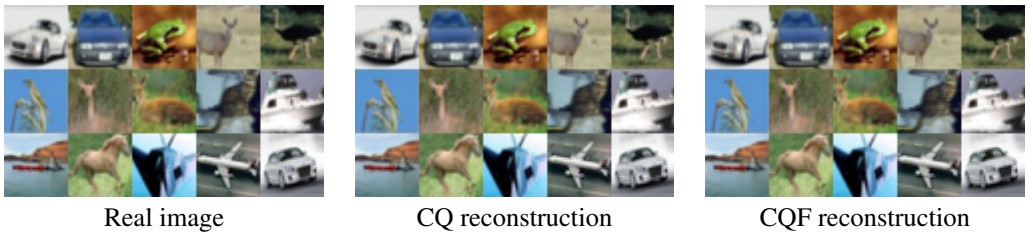

Real image        CQ reconstruction        CQF reconstruction

Figure 7: Real images and their reconstructions with the CQ and CQF models.

