# OpenReview forum: "Coverage and Quality Driven Training of Generative Image Models"
_ICLR.cc/2019/Conference_

### Official Review · AnonReviewer1 · 2018-11-05
**see below**

**Rating:** 7
**Confidence:** 4

**Review:**

The paper presents the use of an invertible transformtion layer in addition to the conventional variational autoencoder to map samples from decoder to image space, and shows it improves over both synthesis quality and diversity.
The paper is well motivated, and the main motivation is nicely presented in Fig.1, and the main idea clearly shown in Fig.2 in an easy-to-understand manner. Existing works are properly discussed in the context before and after the main method. Convincing results are presented in the experimental section, with ablation tests in Tables 1-3, quantitative comparison in Table 4, and qualitative visual images in Figs.4-5.

I incline to my current score after reading the response and other reviews.

---

> ### Author Response · Authors · 2018-11-09
> **Author response to reviewer 1**
>
> Thank you for your review and appreciation of our work. We are happy to discuss any questions that you may have during the discussion period.

---

### Official Review · AnonReviewer3 · 2018-11-07
**Interesting, but there are some unclear issues**

**Rating:** 4
**Confidence:** 5

**Review:**

It is well known that optimizing divergences of different directions leads to different learning behaviors - the mode covering behavior of maximum likelihood training and the mode missing behavior of GAN training. This paper makes a good presentation in explaining coverage-driven training and quality-driven training.
Techinically, this paper make two contributions. First, extend VAEs by using deterministic invertible transformation layers to map samples from the decoder to the image space. Second, use the loss Eq. (8) to train the generator.

However, there are some unclear issues.

First, the differences between losses may not fully explain the behavior of GANs [I. J. Goodfellow. NIPS 2016 tutorial: Generative adversarial networks. arXiv:1701.00160, 2017], as also seen from some recent studies. For example, using the sum of two divergences [Chen et al., 2018] does not make a significant improvement.
Also shown in Table 1, CQ performs better than VAE, but is inferior to GAN.
Using the two-term loss Eq. (8) may not be the key for improvement.

Only after adding the additional flow-based layers, CQF outperforms GAN. Therefore, in Table 1, it would be better to include the result of VAE using the additional flow-based layes for ablation study.

Second, it is also not clear from the paper that such VAE using the additional flow-based layes is new or not.

Third, the results are not as good as the state-of-the-art. In Table 4, SN-GANs perform the best in three out of four cases.

Fourth, the model consists of three networks - encoder, decoder and discriminator. Evaluation about the model's inference capability is necessary in addition to showing its generation capability, since it is equipped with a decoder.

Some typos:
P5: L_Q(p*)+L_Q(p*) ?
Table 4: QSF ?

---

> ### Author Response · Authors · 2018-11-09
> **End of author response to reviewer 3**
>
> [Third, the results are not as good as the state-of-the-art. In Table 4, SN-GANs perform the best in three out of four cases.]
>
> We would like to point out that the fourth measure, BPD, is simply unavailable for GAN methods, since they have a degenerate low-dimensional support in the image space, and lack a mechanism to evaluate the density on the support. In particular held-out train data is likely to be off the low dimensional support, and have an infinite negative log-likelihood. This drawback is fixed in likelihood based models (including ours) that have a full support over the image space, and offer exact or approximate likelihood evaluation.
>
> Although you are right that our approach does not beat the state of the art IS and FID numbers on the CIFAR-10 and STL datasets, it does give results that are fairly close (and better in one case). This is the contribution of our paper: showing that we can have (near) state of the art sample quality and a full support and likelihood evaluation on held-out train data.
> For our model  CQF [+Residual, +flow, +large D] (Ours) the comparison is as follows:
> IS CIFAR-10: ours 8.1, state-of-the-art 8.2 (higher is better)
> IS STL : ours 8.6, state-of-the-art 9.1 (higher is better)
> FID CIFAR-10: ours 18.6, state-of-the-art 21.7 (lower is better)
> FID STL : ours 52.7, state-of-the-art 40.1 (lower is better)
>
>
> [Fourth, the model consists of three networks - encoder, decoder and discriminator. Evaluation about the model's inference capability is necessary in addition to showing its generation capability, since it is equipped with a decoder.]
>
> To evaluate our model’s inference capability, we will include reconstructions of encoded images in the revision of our paper. As is typical for flexible VAEs (IAF for instance) they are indistinguishable from the corresponding ground truth images to the naked eye, and testify outstanding inference performance.

---

> ### Author Response · Authors · 2018-11-09
> **Author response to reviewer 3**
>
> Thanks for your review. We address the main points of this review below and are happy to discuss further if some points are unclear. We will shortly post a revised version of our paper taking into account this feedback.
>
> Our main contribution is a model that can be evaluated in terms of likelihood while also producing compelling samples. Our approach addresses the main drawbacks of existing generative models: For GANs: mode-dropping and the lack of quantitative measures to assess the model fit to held out data. For likelihood-based models: the inferior sample quality as compared to GANs.
>
>
> [Also shown in Table 1, CQ performs better than VAE, but is inferior to GAN.
> Using the two-term loss Eq. (8) may not be the key for improvement.]
>
>  In Table 1, the VAE, GAN and CQ generators have the same architecture and capacity. The use of Eq(8) forces CQ to cover the full support of the dataset. In contrast, a GAN can focus on a subset of the support, and offer better quality on this subset alone. Thus, for a fixed model capacity, the use of Eq (8) is expected to make CQ inferior to GAN in terms of quality (IS, FID), and superior in terms of coverage (BPD). This is indeed what our results show in practice. In this setting, Eq (8) is the only key to improvement in terms of quality over VAE, and in terms of coverage over GAN.
>
>
> [Only after adding the additional flow-based layers, CQF outperforms GAN. Therefore, in Table 1, it would be better to include the result of VAE using the additional flow-based layers for ablation study.]
>
> The performance of a VAE with NVP flow layers trained by MLE is already in the paper, in Table 3b (where we evaluated different configurations of the NVP layers). We agree with your suggestion that it is useful to already present one setting in Table 1, and will do so in our  revision. The results show that with the additional NVP layers, the model is better in all three measures as compared to the baseline VAE. As compared to CQF,  the model trained with MLE is better in BPD, and worse in IS and FID, as expected. As compared to the standard VAE, IS grows by 1.0 point from 2.0 to 3.0 IS, and FID decreases from 171.0 to 112.0. These improved values are still far from typical GAN performance. Samples in appendix B also show over-generalization typical of MLE. This shows, qualitatively this time, that flow alone is not enough, and the adversarial loss in  Eq(8) is essential. CQF then experimentally demonstrates that the key to achieving full performance is to use the two together.
> We will clarify this in the main text, by adding more qualitative examples.
>
>
> [Second, it is also not clear from the paper that such VAE using the additional flow-based layes is new or not.]
>
> To the best of our knowledge, we are the first to propose and report experimental results of a VAE model with NVP layers in the decoder. The NVP decoder layers remove the naive factorized Gaussian assumption typical in VAE decoders, and significantly improves data log-likelihood (BPD) and sample quality. We will underline this in the introduction of the paper.
>
> The third and fourth questions are adressed in the next response.

---

> ### Comment · AnonReviewer3 · 2018-11-27
> **The paper is improved but still some problems**
>
> The reviewer would like to thank the authors for their response, which clarifies some unclear issues.
>
> Still some problems, after reading the updated paper and the response:
>
> Q1: [Also shown in Table 1, CQ performs better than VAE, but is inferior to GAN.
> Using the two-term loss Eq. (8) may not be the key for improvement.]
>
> Authors> In Table 1, the VAE, GAN and CQ generators have the same architecture and capacity. The use of Eq(8) forces CQ to cover the full support of the dataset. In contrast, a GAN can focus on a subset of the support, and offer better quality on this subset alone. Thus, for a fixed model capacity, the use of Eq (8) is expected to make CQ inferior to GAN in terms of quality (IS, FID), and superior in terms of coverage (BPD). This is indeed what our results show in practice. In this setting, Eq (8) is the only key to improvement in terms of quality over VAE, and in terms of coverage over GAN.
>
> This explanation is not convincing. A point that the authors may not be aware of is that we can not say IS/FID only measures quality. The definition of IS includes two terms - the entropy of the conditional label distribution of samples and the entropy of the marginal label distribution. They are intended to measure  the quality (generating realistic samples) and coverage (generating varied samples) respectively. Additionally, FID can detect mode dropping. Therefore, IS/FID reflect both quality and coverage.
>
> For a fixed model capacity, CQ should be better than GAN, if CQ enjoys the merits of both approaches.
>
> However, compared with GAN, CQ is worse in both IS and FID. This means that using the two-term loss Eq. (8) alone does not leverage both coverage and quality driven training. It sacrifice one for the other. Only after adding the additional flow-based layers, CQF outperforms GAN in (IS, FID). From this perspective, the major claim of unifying coverage and quality driven training is not supported by the experiments.
>
> The paper consists of interesting efforts. I would like to suggest the authors to present around the claim - extending GANs with likelihood evaluations or extending VAEs with flow-based output layers and GAN regularization.
>
> Other issues:
> In the key equation Eq.8, we should have <=, instead of >=
>
> In Eq. 4, the right square bracket is wrongly placed.

---

> > ### Author Response · Authors · 2018-11-28
> > **End of author response to additional comments**
> >
> > We now argue that  IS and FID correlate predominantly with the quality of samples. In the literature (mostly the generative adversarial networks literature, for instance Miyato et al. 2018) they are considered to correlate well with human judgement of quality. Empirically, state-of-the art likelihood-based models have very low IS/FID scores despite having good coverage, so the low quality of their samples dominates. Conversely, state-of-the art adversarial models have high IS/FID scores despite suffering from mode dropping (which strongly degrades BPD), so the high quality of their samples dominate. This is especially true when identical architectures and training budget are considered, as in our first experiment. This observation does not yield an estimate of the relative weights of each factor. To obtain a more quantitative estimation of how much entropy/coverage impact IS/FID, we proceed like so: we measure the scores obtained by random subsamples of the dataset, such that the quality is unchanged but coverage progressively degraded (see details of the scores below). When using the full set (50k) images the FID is 0 as the distributions are identical. Notice that as the number of images decreases, IS is very stable (it can even increase, but by very low increments that fall below statistical noise, with a typical std on IS of 0.1). This is because the entropy of the distribution is not strongly impacted by subsampling, even though coverage is. FID is more sensitive, as it behaves more like a measure of coverage (it compares the two distributions). Still, the variations remain extremely low even when dropping most of the dataset. For instance, when removing 80% of the dataset (i.e., using 10K images), FID is at 2.10, to be compared with typical GAN/CQF values that are around 20. These measurement demonstrate that IS and FID scores are heavily dominated by the quality of images. From this, we conclude that IS and FID can be used as reasonable proxys to asses sample quality, even though they are also influenced by coverage a little. We will include this discussion in the appendix.
> >
> > Let us consider the effect of mode-dropping on IS/FID using random subsamples of real CIFAR10 train images.
> > 50K (full train): IS=11.3411, FID=0.00
> > 40K: IS=11.3388, FID=0.13
> > 30K: IS=11.3515, FID=0.35
> > 20K: IS=11.3458, FID=0.79
> > 10K: IS=11.3219, FID=2.10
> > 5K: IS=11.2108, FID=4.82
> > 2.5K: IS=11.0446, FID=10.48
> >
> >
> > > [Other issues: In the key equation Eq.8, we should have <=, instead of >= . In Eq. 4, the right square bracket is wrongly placed]
> >
> > Thank you for signaling these. The inequality in Eq (6) is where the error is: the ELBO is a lower bound on the likelihood, and there is a minus sign, so it has to be inverted.

---

> > > ### Comment · AnonReviewer3 · 2018-11-29
> > > **A worse problem**
> > >
> > > The reviewer would like to thank the authors for their thoughts.
> > >
> > > The claims w.r.t. quality and coverage via IS/FID/BPD is complicated. So the authors would better revise their claim, as suggested in my previous post. This paper need significant re-writing.
> > >
> > > Moreover, the key Eq. 8 is problematic, as I explain below.
> > >
> > > > Thank you for signaling these. The inequality in Eq (6) is where the error is: the ELBO is a lower bound on the likelihood, and there is a minus sign, so it has to be inverted.
> > >
> > > Thanks for pointing that it is Eq. 6 that should be inverted. You are right. But unfortunately, this leads to a worse problem.
> > >
> > > Basically, you can not assume L_Q(p_theta) = D_KL(p_theta||p*), since in practice discriminator optimality is never achieved. We only have L_Q(p_theta) <= D_KL(p_theta||p*).
> > > [I would suggest reading the f-GAN paper which might be helpful to understand this.]
> > >
> > > Therefore Eq. 8 does not hold.

---

> > > > ### Author Response · Authors · 2018-11-29
> > > > **Author response**
> > > >
> > > > We would like to thank reviewer 3 for the continued discussion. We adress the new point raised below, and welcome further discussion.
> > > >
> > > > > [The claims w.r.t. quality and coverage via IS/FID/BPD is complicated. So the authors would better revise their claim, as suggested in my previous post. This paper need significant re-writing.]
> > > >
> > > > It is unclear from this answer what exactly fails to be convincing in our previous response, which was meant to address exactly this. Could you please elaborate? In particular, in light of our previous answer, we strongly believe that only minor rephrasing is required with respect to claims regarding CQF that rely on IS/FID. Given that this point was raised after the paper revision deadline, we did not have a chance to do this rephrasing, but agree that it is necessary and promise to do so.
> > > >
> > > > > Basically, you can not assume L_Q(p_theta) = D_KL(p_theta||p*) [...] Therefore Eq. 8 does not hold.
> > > >
> > > > The equations are written precisely under this optimality assumption. Under this assumption, Eq. 8 does hold.
> > > > We do not claim that this optimality is reached in practice, and agree that it is not. Is is only an approximation, with no way of assessing how precise. We also clarify how the f-GAN paper relates to Eq.8 in the next paragraph, as it is more technical . Note that giving the loss used under optimality assumption is standard practice in the generative adversarial network literature. The fact that the bound is only approximate does not impact our ability to evaluate the likelihood, and $L_Q$ plays the intended role of biasing the model towards mode-seeking behaviour, as is the case with GANs.
> > > >
> > > > > [I would suggest reading the f-GAN paper which might be helpful to understand this.]
> > > >
> > > > Thank you for pointing this insightful paper out. We in fact did study the f-GAN paper. The math that it contains can sometimes be misleading because of how generic the class of Divergence being considered is. We will now discuss it in more detail.  The inequality (4) developed in section 2.2, in the f-GAN paper, can seem problematic, as it may appear that a lower bound and an upper bound are being summed. In fact it is not a problem in our setup. Indeed (4) is an inequality only because it is presented for a much more general set of divergences, and it is an equality when $D_f$ is the reverse Kullback-Liebler. The inequality is not due to the lack of an optimality assumption. Indeed it is obtained _under_ optimality assumption, as it is a supremum over the class of function $\Tau$. The inequality comes from the lack of an _expressivity_ assumption, which is the assumption that the Fenchel conjugate can be achieved under optimality assumption. This distinction between expressivity and optimality assumptions is often not made, because in most standard GAN formulations, as in our setup, this is not an issue: the Fenchel conjugate is attained under optimality assumption. Indeed, the next paragraph in section 2.2 still, details that this bound can  be tight if $f$ is well chosen. In particular, In the case where $D_f(P||Q) = D_{KL}(Q||P)$, which is what we use, one should take  $f= - log(u)$.  With this, the bound is an equality. Therefore, under optimality assumption $L_C + L_Q$ is an upper bound, and Eq(8) holds.

---

> > > > > ### Comment · AnonReviewer3 · 2018-12-01
> > > > > **Still unclear**
> > > > >
> > > > > The reviewer would like to thank the authors for their response.
> > > > >
> > > > > The last paragraph in the above response seems to be written in a rush.
> > > > >
> > > > > > Note that giving the loss used under optimality assumption is standard practice in the generative adversarial network literature.
> > > > >
> > > > > Checking the converged loss when the algorithm is supposed to converge to fixed points is necessary, but does not justify an optimization problem to be well behaved. We need to examine the objective functions used in optimization. Note that in practice discriminator optimality is never achieved, thus we have
> > > > > L_Q(p_theta) <= D_KL(p_theta||p*).
> > > > > And we have
> > > > > L_C(p_theta) >= D_KL(p*||p_theta) + H(p*).
> > > > >
> > > > > Thus the proposed method actually minimizes the sum of an upper bound and a lower bound. To the best of my knowledge, there are no such practices in GAN literature. It is difficult to understand why such practice works well. I suggest the authors write down the objective functions and show what the proposed algorithm converges to.
> > > > >
> > > > > > Indeed (4) is an inequality only because it is presented for a much more general set of divergences, and it is an equality when $D_f$ is the reverse Kullback-Liebler.
> > > > >
> > > > > Indeed (4) is an inequality because the class of functions T contains only a subset of all possible functions, no matter what f is used.
> > > > >
> > > > > > Indeed, the next paragraph in section 2.2 still, details that this bound can  be tight if $f$ is well chosen.
> > > > >
> > > > > if $T$ is well chosen.
> > > > >
> > > > > > In particular, In the case where $D_f(P||Q) = D_{KL}(Q||P)$, which is what we use, one should take  $f= - log(u)$.  With this, the bound is an equality.
> > > > >
> > > > > Taking $f= - log(u)$ gives $D_f(P||Q) = D_{KL}(Q||P)$, but still yields the equality.

---

> > > > > > ### Author Response · Authors · 2018-12-03
> > > > > > **Answer to reviewer 3**
> > > > > >
> > > > > > The authors would like to thank reviewer 3 for his thoughts, and fast answers.
> > > > > >
> > > > > > The last paragraph of our previous response details the validity of Eq(8) under optimal discriminator assumption. It seemed necessary, as in the f-gan paper (https://arxiv.org/pdf/1606.00709.pdf), in section 2.2 Eq(4), a lower bound is obtained even under optimality assumption.
> > > > > > In short, that is because the expressivity assumption is lacking (the assumption that the Fenchel conjugate can be attained when the best $T$ in $\Tau$, $\T^*$ is attained). Assuming $T^*$ is attained, the tightness of the inequality depends on whether the condition presented in Eq(5) can be satisfied or not. This yields constraints on $\Tau$ and on $f$.
> > > > > > We hope this clarifies our previous response. In any case, we seem to agree that in our setting that bound is tight  under optimal discriminator assumption, and so summing $L_Q$ and $L_C$ yields a valid upper bound in theory.
> > > > > >
> > > > > > Let us now address the practical case. How can our loss be used?
> > > > > > 	i) To report the log-likelihood performance of our model, $L_C$ needs to be a valid upper-bound of $D_KL(p^* || q) + H(p)$. That is true in our setting, and so BPD can be reported.
> > > > > > 	ii) $L_Q$ is not an upper-bound of $D_KL(q || p^*)$ in practice, so we cannot report values of the loss in Eq(8) as being measures of performance of our model. This is the same as in the GAN framework: the scores attributed by the discriminator can be arbitrarily wrong if training fails.
> > > > > > 	iii) While the values for the loss in Eq(8) cannot be reported as reliable metrics, they are useful for training the generative model, as we detail below.
> > > > > >
> > > > > > > [Thus the proposed method actually minimizes the sum of an upper bound and a lower bound.
> > > > > > To the best of my knowledge, there are no such practices in GAN literature.
> > > > > > It is difficult to understand why such practice works well. [...]]
> > > > > >
> > > > > > The training consists of two phases, similar to those of GAN training.
> > > > > >  The first phase, discriminator training, brings $L_Q$ closer to the true reverse KL.
> > > > > > The second phase, generator training, minimises $L_Q + L_C$.
> > > > > > The first phase is not impacted by the use of $L_C$ and is identical to the GAN case. In the second phase, outside of the optimal discriminator assumption, $L_Q$ is not an upper-bound on the reverse KL. Therefore minimizing it offers no theoretical guarantees that $D_KL(q || p^*)$ will decrease. Minimizing $L_Q$ is useful only if we assume that the discriminator has learned something about the reverse KL, such that minimizing it will decrease the reverse KL. The GAN literature has shown that to be true in practice. We also experimentally show that this is still true when adding the likelihood term in our setting.
> > > > > > $L_C$, on the other hand, is a valid bound so offers guarantees on how $D_KL(p^* || q)$ decreases. This procedure can be seen as training a GAN, with an additional term that pushes the generator to cover all modes of the data (in a measurable way).
> > > > > >
> > > > > > We hope this clarifies why the training procedure is expected to work in practice and welcome further discussion. Were previously raised issues satisfyingly addressed?
> > > > > >
> > > > > > Thank you,
> > > > > > The authors.

---

> > ### Author Response · Authors · 2018-11-28
> > **Author response to Reviewer 3's additional comments**
> >
> > The authors would like to thank the reviewer for his answer and additional suggestions. We answer the issue raised in the comment below, and welcome further discussion.
> >
> > > [This explanation is not convincing. A point that the authors may not be aware of is that we can not say IS/FID only measures quality. The definition of IS includes two terms - the entropy of the conditional label distribution of samples and the entropy of the marginal label distribution. They are intended to measure  the quality (generating realistic samples) and coverage (generating varied samples) respectively. Additionally, FID can detect mode dropping. Therefore, IS/FID reflect both quality and coverage. ]
> >
> > Thank you for raising this interesting point. IS and FID scores are indeed imperfect proxies to evaluate quality, in part because they both evaluate quality and coverage. To be precise, IS rewards high entropy in the learned distribution, but not exactly coverage in the sense that generated and real images are evaluated separately rather than compared, whereas FID compares the two distributions. Thus, 2 models can have the same IS/FID scores but a different trade-off. Unfortunately, using a metric that measures _only quality_ is not possible, as it would be maximized by always outputting the mode of the true data distribution. An entropy term is thus required for the metrics to not be degenerate. This makes the simple reasoning 'better IS / FID => better quality' impossible, at least in theory, and complicates evaluation. If one assumes that IS and FID are good proxies for quality (or in other words, that they correlate much more with quality than with coverage), this becomes possible again.
> > As we detail below, many of the claims made in our paper do not rely on this assumption. Some claims do rely on it. We argue here that this is a reasonable assumption in practice, and thus our empirical evaluation is sound.
> > We thank the reviewer for signaling that this issue can and should be better addressed in the paper.
> > In particular we agree that claims about CQF should be rephrased to make it clear that improved IS / FID may in part stem from improved coverage.  Therefore for a given fixed value of IS/FID, slightly degraded quality can  be expected for CQF as compared to a GAN with the same score IS/FID scores. While we can not update the pdf at this time, we promise to do so when again possible.
> >
> > We now restate which claims we make and why, and which claims we can not make, this time with and emphasis on how the point above impacts reasoning. The reliable BPD metric shows improved coverage for CQ over the GAN, which is our first claim. IS and FID are significantly improved for CQ over VAE. Given that the BPD is the same for CQ and VAE, the improvement in IS/FID scores  is expected to come from their quality component, so the assumption that IS and FID correlate mostly with quality is not necessary here. Visual inspection also confirms the very significant qualitative improvement over the VAE, hence our second claim. What we do not claim is that our CQ model has improved coverage _and_ quality over the GAN. If a model had improved coverage and the same quality, IS would indeed improve thanks to the entropy term and enable us to make that claim without even looking at the BPD, but experimentally that is not the case. Indeed the IS and FID of CQ degrade somewhat w.r.t. GAN. The fact that this claim cannot be made is not unexpected, as the model capacity is fixed: covering more points of the dataset makes covering a given point well harder, so if IS and FID are assumed to predominantly correlate with quality, they should degrade. As formulated in the review, CQ sacrifices one for the other.
> >
> > As regards the claims made about CQF: as BPD shows, coverage improves. This improvement
> > alone could, in theory, be responsible for the improvement in terms of IS/FID with respect to CQ, and could in theory even compensate a further degradation of quality (still compared to CQ). In practice that is not the case, though it is now clear that some claims must be rephrased to resolve ambiguity in that respect. Indeed IS and FID correlate very predominantly with quality (see below for arguments). Visual inspection also shows significant qualitative improvement from CQ to CQF, and no real degradation from GAN to CQF. To substantiate this so far subjective claim, we will conduct a user study and report the results here and in an updated version of the paper. This is the only way to obtain a measurement that is not impacted by the entropy or coverage. We will also include more samples in appendix so that the reader can verify it himself. In practice most of the improvement in IS/FID comes from improved quality, hence the claim that we can improve coverage at a low cost in terms of quality. We will clarify those claims where appropriate.
> >
> > The end of this response can be found in the next message.

---

### Official Review · AnonReviewer4 · 2018-11-21
**Additional Review**

**Rating:** 5
**Confidence:** 4

**Review:**


The paper proposes to alleviate two issues with VAEs and GANs
Mode covering behaviour of the MLE loss used in VAEs causing blurry image samples despite high likelihood scores (Coverage driven training in the paper)
Poor likelihood scores in GAN based models despite great looking amples (Quality driven training in the paper)

Both of these issues are well known and have been previously described in the literature (e.g. MLE models: bishop 2006, GANs: arjovsky et. al. 2017 or sonderby et. al. 2017)

The main contribution of the paper are described in the Eq (8) augmenting the VAE ELBO (L_C term) with a GAN-discriminator based loss (L_Q term). Combining the fact that the L_Q loss (essentially MLE) minimises KL[data|model] and L_Q discriminator loss used minimises KL[model|data] the authors show that this in theory minimizes a lower bound on the forward and reverse KL divergence i.e. somewhere between mode-covering and mode-seeking behavior

As a secondary contribution the authors show that adding a flow based module to the generative model p(x|z) increases both the likelihood and the fidelity of the samples.
The experimental section seems sound with extensive results on CIFAR10 and some additional results on STL, LSUN, CelebA and ImageNet.

Comments
Overall i find the paper well written and easy to understand and the experiments seems sound. My main criticism of the paper is the novelty of the proposed model. Adding a VAE and a GAN has been done before (e.g. Larsen 2016 as the authors cite as well). Outside of generative models the combination of MLE and GAN based losses have been studied in e.g. Super-Resolution (Shi et. al. 2016). In both papers of these papers the GAN based losses are added for exactly the same reasons as provided in this work i.e. to increase the sample fidelity.

I’m unsure if adding a flow to the output of the VAE generative model have been done before, however in spirit the approach is quite similar in the ideas in “Variational Lossy Autoencoder” (Chen 2016) or PixelVAEs (Gulrajani 2016) where local covariance is added through conditional factorization.

Questions
Q1) One criticism of the results is that the authors claim that the model achieves likelihood values competitive with state-of-the-art where the results seems to suggest that a more sound conclusion is that the the likelihoods are better than GANs but worse than MLE models such as PixelCNN++. Similarly for the Inception/FID scores where the model is better than VAEs but most of the time slightly worse than pure GAN models. ?

Q2) I find the claim “we propose a unified training approach that leverages coverage and quality based criteria” a bit overreaching. The loss in Eq(8) is simply the sum of a VAE and a GAN based loss and as such does not provide any unification of the two ?

Q3) Related to the previous question The KL-divergence interpretation of the GAN based loss assumes a bayes optimal discriminator. In practice this is usually never achieved why it is not really known what divergence GANs minimize if any (see e.g Fedus et. al. 2017 for a non-divergence minimization view). If the KL-divergenve minimization interpretation does not hold the proposed model is essentially a VAE with an auxiliary GAN loss biasing the model towards mode-seeking training?

Q4) There haven’t been many successful GAN usages outside of of the CNN-realm which suggests that a successful GAN is tightly connected to the inductive bias provided by CNNs for images. Have the authors tried there model on something else than images? (I suspect that the GAN based L_Q loss will be hard to apply outside of the realm of images)

Overall i find the paper well written and easy to understand and the experiments seems sound.  However I think that closely related variants of the main contributions in the paper have been tried elsewhere which somewhat reduces the novelty of the proposed model. Given my comments above I think the paper is marginally below acceptance however I could be be convinced otherwise by e.g. a solid use-case for models like this?

Lastly, I'm sorry for the late review (I was called upon late to review the paper) - I hope that you will find the time for a proper rebuttal to my questions.

---

> ### Author Response · Authors · 2018-11-21
> **End of Author response to reviewer 4**
>
> [Q2) I find the claim “we propose a unified training approach that leverages coverage and quality based criteria” a bit overreaching. The loss in Eq(8) is simply the sum of a VAE and a GAN based loss and as such does not provide any unification of the two ?]
>
> Regarding the loss in Eq(8): it is indeed a sum, and perhaps not as 'unifying' as expected. We will tone down the wording in this respect (using for example “combining”?), to avoid any deception here. To effectively benefit from this joint objective function, and obtain both good samples and likelihood scores, we additionally propose an adapted architecture leveraging inverse-autoregressive flow and top-down sampling (Kingma, NIPS’16) and invertible NVP (Dinh, ICLR’17) layers to obtain a non-factorial non-Gaussian decoder. Without this, optimizing Eq(8) does not lead to models that have both compelling samples and good likelihood and in this sense, our proposed approach unifies the two.
>
> [Q3) Related to the previous question The KL-divergence interpretation of the GAN based loss assumes a bayes optimal discriminator. In practice this is usually never achieved why it is not really known what divergence GANs minimize if any (see e.g Fedus et. al. 2017 for a non-divergence minimization view). If the KL-divergence minimization interpretation does not hold the proposed model is essentially a VAE with an auxiliary GAN loss biasing the model towards mode-seeking training?]
>
> Agreed: in practice optimality is never achieved, and worse: there is no way of knowing to what extent. The KL view is an 'idealized' view of the loss. In practice though, what we have is indeed a mode covering mechanism combined with a mode-seeking training mechanism. Although the quality of the reverse KL approximation is unknown, the GAN literature has shown beyond any doubt that it is a powerful technique to achieve mode-seeking training.
>
> [Q4) There haven’t been many successful GAN usages outside of of the CNN-realm which suggests that a successful GAN is tightly connected to the inductive bias provided by CNNs for images. Have the authors tried there model on something else than images? (I suspect that the GAN based L_Q loss will be hard to apply outside of the realm of images)]
>
> Thanks for your comment. We have not tried it outside of image domains. The inductive bias of convolutional layers indeed seems connected to GAN success, and so it is not clear that L_Q could be easily applied the way we do it in other domains. We will study this further in the context of NLP in the future. For example, (Wu et al, “Adversarial Neural Machine Translation”) have explored adversarial training for machine translation.
>
> [Overall i find the paper well written and easy to understand and the experiments seems sound.  However I think that closely related variants of the main contributions in the paper have been tried elsewhere which somewhat reduces the novelty of the proposed model. Given my comments above I think the paper is marginally below acceptance however I could be be convinced otherwise by e.g. a solid use-case for models like this?]
>
> In applications where GANs are typically used (for instance popular ‘photoshop++’ applications that automatically make people look older, or bald) our approach could be used to objectively select the model that best covers the full dataset, with little cost in terms of quality. This would improve performance in otherwise mode-dropped regions of the support, and fixes the lack of an objective, reliable measure to guide developpement choices. In applications where MLE models are used because the compression metric is needed, our model would provide samples of strikingly greater quality, though for now at a cost in terms of BPD.
> We insist also that we are the first to report both BPD and IS+FID measurements on a collection of seven common datasets. We sincerely hope that this provides the basis for more complete evaluation of generative image models in the future.

---

> > ### Comment · AnonReviewer4 · 2018-11-21
> > **Comment 2**
> >
> > >Regarding the loss in Eq(8): it is indeed a sum, and perhaps not as 'unifying' as expected. We will tone down the wording in this respect (using for example “combining”?), to avoid any deception here. To effectively benefit from this joint objective function, and obtain both good samples and likelihood scores, we additionally propose an adapted architecture leveraging inverse-autoregressive flow and top-down sampling (Kingma, NIPS’16) and invertible NVP (Dinh, ICLR’17) layers to obtain a non-factorial non-Gaussian decoder. Without this, optimizing Eq(8) does not lead to models that have both compelling samples and good likelihood and in this sense, our proposed approach unifies the two.
> >
> > Yes I think "combine" describes the approach well
> >
> > >In applications where GANs are typically used (for instance popular ‘photoshop++’ applications that automatically make people look older, or bald) our approach could be used to objectively select the model that best covers the full dataset, with little cost in terms of quality. This would improve performance in otherwise mode-dropped regions of the support, and fixes the lack of an objective, reliable measure to guide developpement choices. In applications where MLE models are used because the compression metric is needed, our model would provide samples of strikingly greater quality, though for now at a cost in terms of BPD.
> >
> >  I agree your model can be used to tune the tradeoff between BDP and IS+FID. However i think that many of these use-cases for image exactly want mode dropping behaviour i.e. output something with high probability under the "image prior" and does not care so much about coverage /entropy of the model distribution ?
> >
> > >We insist also that we are the first to report both BPD and IS+FID measurements on a collection of seven common datasets. We sincerely hope that this provides the basis for more complete evaluation of generative image models in the future.
> >
> > Yes i think this can be a valuable resource.

---

> > > ### Author Response · Authors · 2018-11-23
> > > **Author response to Comment 2**
> > >
> > > [I agree your model can be used to tune the tradeoff between BDP and IS+FID. However i think that many of these use-cases for image exactly want mode dropping behaviour i.e. output something with high probability under the "image prior" and does not care so much about coverage /entropy of the model distribution ?]
> > >
> > > Indeed, the type of applications that we mentioned care first and foremost about giving an output of high quality, and the point of highest likelihood under the true data distribution is the most satisfactory in that respect. Still, when training this kind of algorithm on a rich dataset (say, for our Photoshop++ example, one that contains many different styles of moustaches), it is desirable that the model learns to output all styles of moustaches rather than one. Especially if these styles all have similar likelihood under the true data distribution, as this will yield a richer application at no cost in quality.
> > >
> > > We have also thought about applications where better coverage of the distribution is more crucial, and will discuss them now.
> > >
> > > GANs have recently been used as a source of data augmentation to train classification/segmentation network. In this setup fighting mode dropping (without degrading the quality of samples) is important, as it will improve the positive effects of GAN data augmentation. While this approach it is not particularly helpful when data is abundant, it received a lot of attention in medical imaging (where data is typically scarce). For instance one can refer to [GAN-based Synthetic Medical Image Augmentation for increased CNN Performance in Liver Lesion Classification, https://arxiv.org/abs/1803.01229 (Neurocomputing, 2018)] for an instance of such an application, or [GANs for Medical Image Analysis, https://arxiv.org/abs/1809.06222] for a recent review.
> > >
> > > An other domain in which data is hard to get and this technique has proven useful is astrophysics. In [Enabling Dark Energy Science with Deep Generative Models of Galaxy Images, https://arxiv.org/abs/1609.05796 (AAAI, 2017)] VAEs and GANs are compared over their ability to produce synthetic data in order to improve the calibration of telescopes. They conclude that VAE-generated samples are too blurry and GAN-generated samples suffer from mode-dropping too much (while having superior quality). It is a good example where full dataset coverage and high quality samples are equally important.
> > >
> > > This type of data augmentation also helps in domains closer to core computer vision, when data is imbalanced. For example that is the case in facial expression classification (where certain emotions occur much less frequently), as demonstrated in [Data Augmentation in Emotion Classification Using Generative Adversarial Networks, https://arxiv.org/abs/1711.00648 (PAKDD, 2018)]

---

> > > > ### Comment · AnonReviewer4 · 2018-11-25
> > > > **Reviewer Response to Comment 2**
> > > >
> > > > Yes i agree that the dataaugmentation use-case is a case where something an approach like the one presented here could be useful. Domain adaptation might also be a related useful application.
> > > >
> > > > I think the paper would be a lot stringer if you could show a small experiment from potential use-case, like redoing the experiment from the "Enabling Dark Energy Science with Deep Generative Models of Galaxy Images" paper using your new loss and hopefully show that i worked better. I know this is not feasible given how late my review where posted though.

---

> > > > > ### Author Response · Authors · 2018-11-29
> > > > > **Thanks**
> > > > >
> > > > > The authors would like to thank Reviewer 4 for all the constructive feedback and suggestions for improvement provided during this open review process.

---

> ### Author Response · Authors · 2018-11-21
> **Author response to Reviewer 4**
>
> Thank you for your review and suggestions. We give our answers to the main points raised below, and welcome further discussion.
>
> [Overall I find the paper well written [...] for exactly the same reasons as provided in this work i.e. to increase the sample fidelity.]
>
> These papers indeed also use both a MLE criterion and an adversarial term. We will better highlight the differences between these works and ours in the related work section. In particular:
> The strongest difference with Larsen (2016) is that they use regression in the feature space of a discriminator. The function that goes from image space to feature space is non invertible and discards information, so a valid BPD is unavailable. Their approach yields an inference mechanism, but not a valid coverage driven procedure: for instance, the discriminator can collapse 2 points X_1 and X_2 in RGB space to a single point Y = f(X_1) = f(X_2) in feature space (paragraph 2.3 of their paper describes this setup). This is why it is crucial for the layers between feature prediction space and image space to be invertible.
> In general, prior work does not provide IS/FID and BPD together, see Table 4. In tables 4 and 5, we provide such results on seven datasets commonly used to evaluate generative image models. These results provide a basis for more complete experimental comparisons for future research in this line of work.
> The introduction of our flow-based decoder architecture is crucial to achieve both high sample quality and low bits-per-dim scores, as experimentally shown in Table 1, and is a novel contribution.
>
> [I’m unsure if adding a flow to the output of the VAE generative model have been done before, however in spirit the approach is quite similar in the ideas in “Variational Lossy Autoencoder” (Chen 2016) or PixelVAEs (Gulrajani 2016) where local covariance is added through conditional factorization.]
>
> Yes, it is a novel contribution. We tried to make that clearer in the revised paper following the remarks of reviewer 2. It is indeed similar in spirit to Lossy VAE and PixelVAEs in its endeavour to model images with non factorial decoders. A crucial difference is that auto-regressive models suffer from slow sequential sampling. This makes it impossible to train PixelVAEs in an adversarial setup, which requires sampling from the model. Our proposed non factorial decoder does not suffer from this issue and can be trained with Eq(8).
>
> [Q1) One criticism of the results is that the authors claim that the model achieves likelihood values competitive with state-of-the-art where the results seems to suggest that a more sound conclusion is that the the likelihoods are better than GANs but worse than MLE models such as PixelCNN++. Similarly for the Inception/FID scores where the model is better than VAEs but most of the time slightly worse than pure GAN models. ?]
>
> Our models indeed do not achieve state of the art likelihood scores, and we will rephrase the claims in the paper accordingly to avoid any confusion in this respect. Our best performing model in Table 4 achieves 3.48 BPD. The best known results were reported for PixelCNN++: 2.92 BPD, but as discussed auto-regressive models are not suitable for adversarial training. Results for VAE-IAF (Kingma et al., 2016), 3.11 BPD, were obtained using an architecture with many more convolutional layers.
> When comparing training strategies using the same architecture, as in Table 1, the BPD values are similar. In particular, VAE (ie MLE training) and CQ obtain both 4.4, while with the additional NVP layers in the decoder results are 3.5 and 3.9 for VAEF and CQF, respectively. For reference, a BPD of 3.5 was reported for NVP (Dinh et al., 2017).
> We recall that, unlike us, none of these papers report a quantitative assessment of sample quality (eg in terms of IS and/or FID).

---

> > ### Comment · AnonReviewer4 · 2018-11-21
> > **Comment 1**
> >
> > >>[I’m unsure if adding a flow to the output of the VAE generative model have been done before, however in spirit the approach is quite similar in the ideas in “Variational Lossy Autoencoder” (Chen 2016) or PixelVAEs (Gulrajani 2016) where local covariance is added through conditional factorization.]
> >
> > >Yes, it is a novel contribution. We tried to make that clearer in the revised paper following the remarks of reviewer 2. It is indeed similar in spirit to Lossy VAE and PixelVAEs in its endeavour to model images with non factorial decoders. A crucial difference is that auto-regressive models suffer from slow sequential sampling. This makes it impossible to train PixelVAEs in an adversarial setup, which requires sampling from the model. Our proposed non factorial decoder does not suffer from this issue and can be trained with Eq(8).
> >
> > Yes i agree it would be a bit cumbersome with adversarial training on samples from autoregressive models.
> >
> > I forgot to mention in my review that i think you should comment on the differences to Rosca et. al. 2017  "Variational Approaches for Auto-Encoding Generative Adversarial Networks". Your model seems like a extension / special-case depending on how you frame it :) of some of the variants considered there´

---

> > > ### Author Response · Authors · 2018-11-23
> > > **Author response to Comment 1**
> > >
> > > [I forgot to mention in my review that i think you should comment on the differences to Rosca et. al. 2017  "Variational Approaches for Auto-Encoding Generative Adversarial Networks". Your model seems like a extension / special-case depending on how you frame it :) of some of the variants considered there´]
> > >
> > > Thank you for pointing out this reference that we did not know, we will add it to the paper.  Our approach indeed fits in their line of work in that they share the motivation of leveraging the VAE and GAN frameworks to get the best of both worlds.
> > >
> > > Their work offers an interesting point of view on motivating this framework. One notable difference with our approach is the choice of loss made for maximum likelihood training. In their work, they note that a discriminator can be used to estimate likelihoods when these are intractable (Paragraph 5 of section 2), an approach that they call Synthetic Likelihood. They then use this it to estimate the KL term between prior and posterior distributions over latent variables in the variational lower bound (see approximation of D_{KL}(q(z|x)||p(z)) with  Eq(6) in section (3) ).
> > >
> > > This choice offers flexibility, as it enables them to do away with the gaussian assumption over latent variables typically made with VAEs. That is possible because they no longer need tractable explicit computations. However, this also means that the loss used is no longer a valid lower bound on the likelihood of the data.  In particular, the latent KL estimation can suffer from mode-dropping, and there is no way of measuring the error made by this approximation. Therefore, they do not have access to a valid Likelihood performance which is a difference with our framework.
> > >
> > >  An other difference that we introduce is the use of the non-factorial decoder. We also applied our methods to datasets of increased resolutions, and with more variability in the data (For instance with STL-10 and Image-net). The best quantitative performances that we report are higher (for instance 8.1 IS for our best model on Cifar-10 VS below 7 (barplots provided) IS for theirs).  The architectures, however, differ and so results are not directly comparable. We will include this work in a revised version of our related work section soon.

---

### Author Response · Authors · 2018-11-16
**Updated version**

We have updated the pdf to take into considerations some remarks made in the review. Specifically,
the ablation study in Table 1 has been updated as suggested, the novelty of the architecture used is stated more clearly, visualisations of reconstructions are available in appendix C, and approximate likelihood performance for the GAN baseline, obtained by training an inference network with a frozen gan generator as decoder, is now provided.

---

### Meta-Review · Area_Chair1 · 2018-12-14
**Not enough for acceptance**

**Confidence:** 5
**Recommendation:** Reject

**Metareview:**

The overall view of the reviewers is that the paper is not quite good enough as it stands. The reviewers also appreciate the contributions so taking the comments into account and resubmit elsewhere is encouraged.